# Mesenchymal Stem Cells in Embryo-Maternal Communication under Healthy Conditions or Viral Infections: Lessons from a Bovine Model

**DOI:** 10.3390/cells11121858

**Published:** 2022-06-07

**Authors:** Alexandra Calle, Miguel Ángel Ramírez

**Affiliations:** Department of Animal Reproduction, National Institute for Agriculture and Food Research and Technology (INIA), CSIC, 28040 Madrid, Spain; calle.alexandra@inia.csic.es

**Keywords:** mesenchymal stromal cells, extracellular vesicles, early pregnancy, embryo, trophectoderm

## Abstract

Bovine mesenchymal stem cells are a relevant cell population found in the maternal reproductive tract that exhibits the immunomodulation capacity required to prevent embryo rejection. The phenotypic plasticity showed by both endometrial mesenchymal stem cells (eMSC) and embryonic trophoblast through mesenchymal to epithelial transition and epithelial to mesenchymal transition, respectively, is essential for embryo implantation. Embryonic trophoblast maintains active crosstalk via EVs and soluble proteins with eMSC and peripheral blood MSC (pbMSC) to ensure the retention of eMSC in case of pregnancy and induce the chemotaxis of pbMSC, critical for successful implantation. Early pregnancy-related proteins and angiogenic markers are detected as cargo in EVs and the soluble fraction of the embryonic trophectoderm secretome. The pattern of protein secretion in trophectoderm-EVs changes depending on their epithelial or mesenchymal phenotype and due to the uptake of MSC EVs. However, the changes in this EV-mediated communication between maternal and embryonic MSC populations infected by viruses that cause abortions in cattle are poorly understood. They are critical in the investigation of reproductive viral pathologies.

## 1. Introduction

Many physiological processes, including embryo development, require effective communication between cells and tissues. Mammalian preimplantation embryos develop in the female genital tract (oviduct and uterus), and effective communication is necessary for their development and survival. It was considered that the bovine embryo was completely autonomous and did not require communication with the mother until day 7–8 of embryonic development [1], converting the oviductal transit of the embryo before it entered the uterus into a simple journey devoid of embryo-maternal communication. However, in 2016 Lopera-Vasquez et al. designed an in vitro model of communication between the bovine embryo and maternal oviductal cells, attributing to their secretome, composed of soluble proteins and extracellular vesicles, fundamental roles in the proper development and growth of embryo during its oviductal transit [2,3]. In other words, while the embryo develops and travels towards the endometrium through the oviduct, the active communication between the two implies optimization of its development. Subsequently, other authors have deepened bovine embryo-maternal communication during the oviductal transit of the embryo [4,5], and findings have been reported in other species of mammals such as the sow and the mare [6]. The embryo in the morula stage (day 5–6) reaches the uterus, where it develops into a blastocyst (day 7–8) and hatches from the zona pellucida (day 8–9) [7]. During this preimplantation period, embryo-maternal communication is essential for correct embryonic development. Failures in this communication are responsible for the high rate of early embryonic loss (40% between days 8 to 17 post-fertilization) in cattle [8].

In addition, as a result of this embryo-maternal communication, an intensive modulation of the immune response in the endometrium is produced, which is essential to prevent embryonic rejection since the embryo constitutes a semi-allogenic “foreign body” (50% of its genetic makeup is of paternal origin) [9]. The presence of the embryo triggers changes in immune cell populations and induces the production of molecules involved in mediating immune tolerance [10]. Immunomodulation is produced by the presence of the embryo in the uterus both locally and systemically in peripheral blood [11]. The immune response has traditionally been classified into two types: Th-1 and Th-2. Even though several human studies have sought to justify pregnancy as a Th-2 or anti-inflammatory state [12], a comparable number of studies have supported its status as a Th-1 or pro-inflammatory state [13]. More recently, a two-phase model, with the first and third trimesters defined by a pro-inflammatory environment and the second trimester representing an anti-inflammatory phase, was suggested [14].

## 2. The Role of Maternal MSC in Embryo Implantation

### 2.1. Evidence for the Existence of MSC in the Maternal Reproductive Tract

Mesenchymal stem cells, also documented as multipotent stromal cells or mesenchymal stromal cells (MSCs) [15], are multipotent cells with notable clinical relevance due to their potential application in cell therapy for regenerative medicine and tissue engineering [16]. Although Prianishnikov hypothesized the presence of stem cells in the endometrium in 1978 [17], it wasn’t until 2004 that stem cells were located and described for the first time in endometrial tissue [18]. The existence of mesenchymal stem cells in the bovine endometrium (eMSC) has also been reported by different authors [19,20,21]. Clonogenicity, ability to attach to plastic, fibroblast shape, and in vitro differentiation potential to adipocytes, chondrocytes and osteocytes are all characteristics of bovine eMSCs during the entire estrous cycle [22,23].

The uterus is a histologically dynamic organ, and the processes that coordinate its regeneration during the estrous cycle and implantation remain unknown. It has been suggested that bone marrow-derived cells aid in uterine renewal [24]. According to Cunha et al., MSCs are an important component of the hematopoietic stem cell niche in the bone marrow [25]. MSCs are a special type of multipotent progenitor that may maintain hematopoiesis while also differentiating into osteogenic, adipogenic, and chondrogenic lineages [26]. MSCs can migrate into damaged tissues from the bone marrow or peripheral circulation, suggesting that eMSCs are probably derived from the bone marrow. It has been documented that between 2% and 52% of the epithelium and endometrial stroma in women who received an HLA-mismatch bone marrow transplant for cancer treatment originated from the transplanted bone marrow [27]. Mints et al. found that around 8% of the uterine epithelium and 9% of the uterine stroma were sourced from bone marrow [28]. Human endometrial endothelial progenitor cells differentiate into endometrial glandular epithelial, stromal, and endothelial cells [29]. Before obtaining blood samples, mobilizing the bone marrow with granulocyte colony-stimulating factors (G-CSF) does not affect red blood cell parameters, but it does increase the number of nucleated cells in the peripheral blood, which includes the pbMSC population [30]. However, the results demonstrate that prior G-CSF mobilization has no significant benefit in MSC isolation in cows. MSC functions were not observed to differ between mobilized and untreated MSCs by Calle et al. [31].

### 2.2. Relevance of Immunoregulation during Implantation

Multiple studies have demonstrated that MSCs are immunosuppressive both in vitro and in vivo, decreasing both B and T cell proliferation [22,30] and reducing the risk of graft-versus-host disease in allografts and xenografts [32]. MSCs have the special ability to move to injured and inflammatory tissues, where they multiply and use their immunomodulatory characteristics as part of the tissue’s regeneration process [33]. MSCs can restrict or limit T cell proliferation, as well as play a role in the control of Th-1/Th-2 balance, transitioning a Th-1 phenotype to a Th-2 phenotype [34]. In the human species, mesenchymal stem cells (MSCs) immunomodulate the maternal immune response [35], preventing abortions [36]. MSCs interactions with immune cells have been extensively researched in humans, rats, and even pigs [33]; however, this study has yet to be extended to MSC from other mammalian species.

The immune system in the endometrium of ruminants, which protects the uterus against infections [37], must also be controlled to prevent embryo rejection [38]. The percentage of diverse populations of endometrial and circulating leukocytes is varied in cattle during mid-and late pregnancy [39], indicating a complicated control in ruminants, as in other animals with an invasive form of implantation.

### 2.3. MSC Plasticity

During the window of implantation, after blastocysts attach to the endometrium in both humans and mice, the stromal cells at the implantation site begin to decidualize [40]. Decidua is identified morphologically as a tissue composed of endometrial fibroblast cells which become rounded or polyhedral as a result of glycogen or lipid buildup in the cytoplasm. These tissular morphological changes can occur during pregnancy, pseudo-pregnancy, or in experimentally generated deciduomas [41]. One of the earliest uterine adaptations to pregnancy in mice and primates is the differentiation of decidual cells [42], which involves cell proliferation, changes in shape, multinucleation, and the formation of intercellular connections [43]. The process of decidualization, which is required for blastocyst implantation and pregnancy maintenance, involves many genes and signal pathways. Decidualization is not shown in species with non-invasive implantation, such as domestic animals. Ruminants and pigs both have non-invasive implantation, but the type of placentation differs. Epitheliochorial placentation occurs in pig conceptuses. The luminal epithelium (LE) is maintained biologically intact throughout pregnancy, and the conceptus trophectoderm only attaches to the apical LE surface without interacting with uterine stromal cells [44]. Synepitheliochorial placentation in ruminants, on the other hand, results in severe LE erosion owing to syncytium development with trophectoderm binucleate cells. Conceptus tissue opposes but does not penetrate uterine stroma after Day 19 of pregnancy. According to Johnson et al., the uterine stroma of sheep undergoes a differentiation process similar to decidualization in invasive implanting species, while porcine stroma shows less differentiation than sheep, rodents, or primates. The degree of uterine stromal decidualization thus varies among species and coincides with the depth of trophoblast invasion during implantation [45]. Few studies have focused on the role of alterations in the uterine stroma that support uterine luminal epithelium since domestic animals have mainly non-invasive implantation.

In situ cellular transdifferentiation, such as mesenchymal to epithelial transition (MET), would represent an alternate or additional mechanism of endometrial regeneration. MET reprogram mesenchymal cells, causing them to lose mesenchymal cell properties while gaining epithelial cell characteristics [46]. MET and the epithelial to mesenchymal transition (EMT) are important processes that occur throughout the development of the bovine embryo and have been correlated to tumor metastasis [47]. According to Uchida et al., the EMT may aid human embryo implantation by allowing the embryo to cross through endometrial epithelial cells and into the endometrial stromal cell layer [48]. MET may also present during decidualization, according to Zhang et al., providing a stable developmental environment and an anchor point for embryos to penetrate the uterus [49]. Patterson et al. revealed that MET aids endometrial regeneration in mice after spontaneous and artificial decidualization using Sesame oil injections into the uterine lumen [50]. However, it has not been investigated whether cellular transdifferentiation could be used to rebuild mature endometrium in bovines. Early and late luteal phase bovine eMSC lines have a fibroblast-like shape and express vimentin. eMSC lines derived from the follicular phase (in the absence of an embryo) express cytokeratin, showing an epithelial-like shape. This finding could indicate eMSC is undergoing a mesenchymal-to-epithelial transition (MET) [30]. Both bone marrow-derived cells [24] and MET of resident cells [50] were involved in uterine regeneration in both mice and humans. In a mouse model of menstruation, Cousins et colleagues established that MET contributes to endometrial epithelium during progesterone withdrawal-induced tissue disintegration [51]. Yu et al. found that luteal hormone concentration and/or duration thresholds induce plasticity and reversibility of endometrial stromal phenotype via MET [52]. The rapid regression of the cow’s corpus luteum is a critical event in the bovine estrous cycle. It is responsible for the abrupt decrease in progesterone levels in the blood, signaling the end of the cycle (follicular phase) [53]. In cows, the findings on differentiation induction of eMSC lines to mesodermal lineage have validated their multipotency in the early and late luteal phases, as well as the follicular phase. eMSC lines derived from various stages of the estrous cycle exhibit a distinct mesenchymal pattern in the early luteal phase (corresponding to the presence of the embryo in the oviduct) and late luteal phase (corresponding to the presence of the embryo in the uterus). While eMSC from the follicular phase display an apparent mesenchymal to epithelial transition state (corresponding to the regression of corpus luteum if there was no embryo), they also express cytokeratin and, on limited occasions, have an epithelial appearance [22]. Given that both the epithelium and the stroma contain estrogen and progesterone receptors during the bovine estrous cycle [54], it would be interesting to investigate the paracrine function of eMSCs in future experiments due to the possible expression of hormone receptors during the estrous cycle.

## 3. The Other Side of the Line: The Embryo Trophoblast

Most research on the bovine embryo–maternal communication has been conducted in vitro, using a trophectoderm primary culture (CT-1) produced by coculture with mouse feeder layers [55]. Through a novel biopsy and culture system that did not require coculture with murine cells, the isolation and characterization of primary cultures of bovine blastocyst embryonic trophectoderm cells (BBT) have been reported. The expression of genes from early trophoblastic markers (*CDX2*, *ELF5*), mononucleated cells (*IFNT*), and binucleated cells (*PAG1*, *PRP1*, and *CSH2*) changed with time in culture, demonstrating that these initial cultures are dynamic populations [56].

It has been shown that the embryonic trophectoderm upregulates the expression of genes associated with an epithelial to mesenchymal transition (EMT) during the embryo’s attachment to the endometrium on day 22 [57]. EMT-related genes, as well as cytokeratin, are found in the bovine TE following the conceptus-endometrium attachment. EMT-related genes (*SNAI2*, *ZEB1*, *ZEB2*, *TWIST1*, *TWIST2*, and *KLF8*), as well as cytokeratin, are found in the bovine trophectoderm cells following the conceptus-endometrium attachment [57]. Lee et al. found that cell-cell communication continues after the conceptus is implanted into the endometrium, and that EMT is a crucial process for numerous processes that help embryonic development, including the formation of basal epidermal compartments [58]. Trophectoderm cells must be more flexible at this time to form binucleated and trinucleated cells. Binucleated trophoblast cells were discovered to have intermediate properties between epithelial and mesenchymal phenotypes [59]. The first type of EMT is thus connected with implantation, embryo formation, organ development, and the creation of several cell types with mesenchymal phenotypes [60].

Following conceptus implantation, the trophectoderm loses the adherence junction molecule, CDH1, and acquires the expression of mesenchymal markers like VIM and CDH2 while still expressing the epithelial marker cytokeratin. On days 20–22, trophectoderm EMT was controlled by the endometrium through activin A and FLST1 [61]. CH1 was found in the cytoplasm of trophoblast binucleate cells in the bovine placentome, and b-catenin translocation into the nucleus was detected [62], demonstrating the role of the CDH1–b-catenin axis in trophoblast differentiation. On day 22, trophoblast CDH2 expression is much higher, suggesting that a rise in CDH1 degradation is responsible for the following drop in its expression. Once it results, as the conceptus connects to the luminal epithelium, CDH1 depletion may play a significant role in the gene expression change required for successful implantation to placentation. The expression of CDH1 was overrepresented in the soluble fraction secreted by trophectoderm primary cultures with epithelial phenotype, while the soluble fraction secreted by trophectoderm primary cultures with mesenchymal phenotype expresses VIM and maintains cytokeratin expression [31].

## 4. Interaction between Embryonic and Maternal MSC in Homeostasis—The Role of the Secretome: Soluble Mediators and Extracellular Vesicles

### 4.1. Trophoblastic-Derived Secretome and EMT

Although the EMT process has been found in bovine trophectoderm cells after embryo implantation [57,61,63], its implications on the cell secretome require still further investigation.

Several growth factor signals, including transforming growth factor (TGF-*β*), hepatocyte growth factor (HGF), epidermal growth factor (EGF), fibroblast growth factor (FGF), Wnt proteins, and IL-6, modulate EMT induction at the molecular level [64]. TGF-*β* is a multifunctional cytokine considered the primary cause of EMT. TGF signaling regulates cell proliferation, differentiation, invasion, migration, apoptosis, and microenvironmental remodeling, as well as inducing pathophysiology EMT and metastasis [65]. TGF-*β* and FGFR1 proteins were highly expressed in the secretome of trophectoderm primary cultures, indicating that they are involved in the regulation of EMT in the bovine trophectoderm [31].

FGF promotes cell mobility by boosting vimentin expression and inducing MMP2 activity during EMT. FGF also alters the actin cytoskeleton, allowing for anchorage-independent growth [66]. Vimentin and MMP2 expression are significantly increased in mesenchymal trophectoderm cells’ EVs and soluble proteins, respectively [31]. Although bovine trophoblasts do not infiltrate the endometrium, MMP2 metalloproteinase overexpression indicates that it may have a function in the non-invasive trophectoderm. FGF1, the ligand for FGFR1, is known to upregulate MMP13 [67], contributing to EMT induction, and is overrepresented in the soluble fraction released by mesenchymal trophectoderm cells [31].

### 4.2. Embryonic Secretome and Inflammatory Cytokines Induce Maternal MSC Chemotaxis towards the Implantation Niche

Maternal hormones control the earliest stages of uterine remodeling for implantation, regardless of whether the embryo is present. A successful pregnancy, on the other hand, requires embryo identification by the mother organism, as well as a crucial contribution to the uterine response. Cows hatch typically 7 to 10 days after fertilization, with the earliest attachments between trophectoderm and endometrium happening until around day 20 of gestation, in contrast to mice and humans, where implantation occurs shortly after hatching [68]. During the brief period between hatching and embryo attachment in humans, a gradient of chemokines and cytokines secreted by endometrial cells has been identified to direct the blastocyst to the implantation site [69]. During the elongation phase, which ends when trophectoderm cells adhere to the luminal endometrial epithelium, bovine trophectoderm cells produce IFN-τ. IFN-τ inhibits luteolysis, which is required to produce progesterone from the corpus luteum [70].

eMSC lines can migrate in the absence of cytokine stimuli. In terms of non-stimulated migration, there was no discernible difference between eMSC lines from different oestrus phases or with distinct morphologies (mesenchymal or epithelial). Although the eMSC lines isolated during the follicular phase (the absence of an embryo) express cytokeratin, and two of these lines displayed apparent epithelial type shape, they also demonstrated a significant migratory capacity and intracellular Vimentin expression [22]. Vimentin is a necessary regulator of mesenchymal cell motility and an important mesenchymal biomarker for epithelial transition. Vimentin protein and mRNA expression are reduced during MET when cell mobility reduces and cells adopt epithelial features [71,72]. Considering the preceding, bovine eMSC exhibit MET cellular flexibility, allowing them to exhibit mesenchymal and epithelial cell properties.

In all scenarios, whether the embryo is still in the oviduct, the embryo without zona pellucida is in the uterus, or the corpus luteum regresses due to the absence of an embryo in the uterus, most of the eMSC presented active migration towards a pro-inflammatory niche (Th-1) characterized by the presence of IL-1β [22] and also described in the early and late stages of human gestation [14]. In contrast, the majority of the eMSC from the three scenarios mentioned respond with a block in migration to IFN-τ [22], the key embryo-derived pregnancy signal in bovines [73].

Under standard conditions, circulating MSCs are found in low quantities in the peripheral blood [74]. Their trafficking, on the other hand, is triggered by an injury, chronic diseases, or cancer [75]. Activated MSCs are released into the peripheral blood circulation [76]. This increase in circulating MSCs is assumed to be sourced from bone marrow, while MSCs from other sources, such as adipose tissues, may also be recruited. Chemokines and homing receptors interact specifically to guide adherence to various locations of damage, cancer, or implantation. In the mouse estrous/menstrual cycle, bone marrow-derived MSCs are implicated in uterine regeneration following injury, but not in cyclic regeneration of the endometrium, according to Du et al. [77].

In cows, the ability of pbMSCs to migrate to an inflammatory environment has also been analyzed. TNF-α does not cause pbMSC to move, although pbMSC lines obtained following G-CSF mobilization from bone marrow did exhibit a reduction in cell migration. The inflammatory cytokine IL-1β, on the other hand, increased the migratory capability of pbMSC recovered with or without bone marrow stimulation [30]. It has been revealed that pbMSC isolated with or without bone marrow stimulation have increased their migratory potential towards the IFN-τ implantation cytokine [30]. These findings contrast markedly with IFN-τ induced reduction of migratory ability in bovine MSC lines generated from the endometrium [22]. Therefore, the embryonic implantation niche would also be a site of active MSC recruitment.

In conclusion, under normal circumstances, bovine MSCs circulate in the peripheral circulation. The fact that inflammation/implantation signals and signals from embryonic trophectoderm induced pbMSC chemotaxis suggests that endometrial mesenchymal stem cells originated in the bone marrow during bovine pregnancy to help keep the immune response low and prevent embryo rejection by the maternal organism. It must be considered that MSCs derived from different tissues have demonstrated heterogeneity based on their migratory capacity [30]. Therefore, to better comprehend MSCs’ status in circulation and maximize their therapeutic potential, it’s essential to characterize their capacity to disseminate and migrate, considering that MSCs originating from diverse tissue sources may have distinct reactions. According to the literature, MSCs demonstrate tissue and donor-related diversity, not only in mRNA expression but also in chemokine and cytokine production [78,79,80,81,82]. The migratory ability of MSC varies depending on the tissue of origin, both in the absence of stimulation and as a result of activation by inflammatory cytokines [83]. In the scenario of pregnancy, the combination of both IL-1β and IFN-τ signals would ensure the retention of eMSC as well as continued recruitment of MSC from the circulation, ensuring the immunomodulation required in the mother for embryo survival (Figure 1). This result supports the concept that either IFN-τ or a Th1 cytokine-like IL-1β could promote the migration of pbMSCs towards the implantation site in the reproductive tract, modulating the Th1 maternal response while retaining eMSC already in the tissue [22].

Later stages of development are not possible in pure in vitro systems, so investigations of embryo-maternal communication have been based on Trophoblastic cells derived from hatched embryos [30,31]. Although the diverse trophectoderm cell lines described have different properties, some of them similar to those seen later in embryo development, it would be fascinating to investigate the chemotactic stimuli released by trophoblastic cells derived from elongated embryos. Trophectoderm cells with a mesenchymal phenotype have developmental characteristics that are closer to EMT, and so would represent a more advanced developmental stage, closer to embryo implantation [31]. A careful analysis was performed, in which the trophectoderm secretome was dissected into soluble mediators and extracellular vesicles (EVs), and the chemotactic activity of both fractions was analyzed independently in parallel. Interestingly, epithelial embryonic trophectoderm lines drive chemotactic migration of maternal eMSCs via both soluble and EVs mediators, while chemoattraction of pbMSCs is induced only via soluble mediators. In contrast, when the embryonic trophectoderm has already developed mesenchymal properties, it induces endometrial or peripheral maternal MSC migration via both soluble and EV-cargo proteins, and these cells can travel greater distances and at faster speeds [31]. Secretome-dependent signaling might thus cause a substantially amplified call effect in MSCs at late implantation stages to guarantee embryo implantation at that key time of pregnancy.

### 4.3. Tissular Rearrangements for Implantation

Following EMT, a micro-angiogenesis process associated with uterine vascularization is required for effective implantation [84] and placenta formation. In the trophectoderm cell secretome, eleven proteins implicated in angiogenesis pathways were identified, six of which participate in the vascular endothelial growth factor (VEGF) pathway [31] (Figure 2). The VEGF signaling pathway is required for all stages and processes of vascular formation (vasculogenesis, angiogenesis, and lymphangiogenesis). VEGF is the primary regulator of angiogenesis in bovine pregnancy. Interestingly, four angiogenic-related factors were overrepresented in the secretome of mesenchymal trophectoderm cells: fibroblast growth factor receptor 1 (FGFR1), RHO GTPase-activating protein 1 (ARHGAP1), RHO-related GTP-binding protein (RHOC), and vascular cell-adhesion molecule (VCAM1). However, only the Serine/threonine-protein kinase A-RAF (ARAF) and Heat Shock Protein Beta-1 (HSPB1) from this pathway were shown to be overrepresented in the secretome of epithelial trophectoderm cells [31]. In conclusion, both the overexpression of EMT markers and the increased detection of angiogenic factors are observed in trophectoderm cell populations at a more advanced embryonic stage near an EMT stage. However, the characteristic bovine pregnancy angiogenic related markers such as VEGF family proteins or its receptor, Angiopoietin (ANGPT)-2/ANGPT-1 [85], were not detected in the secretome of trophectoderm cells, most likely due to an early embryonic stage [31]. In addition, on Day 13 in vivo conceptuses, MMP2, a member of the matrix metallopeptidase family, which is involved in the degradation of extracellular matrix in normal physiological processes [57], and PEG3 are both upregulated [57]. MMP2 and PEG3 are essential implantation proteins whose expression has only been observed in in vivo conceptuses [86].

The expression of integrins (ITGs) at the uteroplacental interface during trophectoderm attachment and placentation has been studied in bovine species [87]. ITG*αV*, overrepresented in Mesenchymal trophectoderm cells-EVs cargo [31], is known to bind to Osteopontin in conjunction with the *β*5 subunits (SPP1) [88]. ITG*β*1 is similarly overrepresented in the cargo of mesenchymal trophectoderm cells-EVs [31]. It can form heterodimers with the 4 chain, also defined as very late antigen-4 (VLA4), commonly found in MSC [89], and the α8 subunit, resulting in alternate Osteopontin SPP1 receptors [90].

Yamakosi et al. discovered that the subunits of SPP1-binding ITGs are increased during embryo attachment, implying a role in trophoblast adhesion to the uterine epithelium in cows [57]. Furthermore, ITG on EVs has been shown to guide organ-specific EVs uptake to induce pre-metastatic niche development in a tumor environment [57]. Similarly, integrin expression profiles of trophectoderm-secreted exosomes may be advantageous for managing maternal MSCs in the implantation niche. They may be considered diagnostic biomarkers to predict successful maternal immunoregulation to prevent embryo rejection. The principal counter ligand of ITGα4β1, vascular cell adhesion molecule-1 (VCAM-1), is critical in leukocyte recruitment during an immunological response [91]. According to Bai et al., uterine VCAM-1 expression was low in cyclic and pregnant animals on day 17 but enhanced between days 20 and 22 of pregnancy [89]. The authors also observed that uterine flushings increased VCAM-1 expression in CT-1 cells (a primary trophoblast culture grown using STO mouse feeder cells) [57]). Additionally, VCAM-1 expression is increased in the cargo of mesenchymal trophectoderm cells-EVs. [31]. Galectin 3 (LGALS3), which also plays an important role in immune system regulation by regulating both innate and adaptive responses in physiological and pathological processes [92], has been linked to the implantation process [93] and is overexpressed in mesenchymal trophectoderm cells-EV cargo [31].

Many invasive, proliferative, and immunological tolerance processes that allow for pregnancy also occur in malignant tumors to promote angiogenesis to ensure nutrition supply and induce an immunologically depressing environment to elude the host’s immune response [94,95,96]. Thus, understanding how all these factors interact during the physiologic process of pregnancy could also aid in the development of new cancer treatment options.

### 4.4. Maternal MSC-EV-Cargo Modulate Embryonic EV-Cargo

After studying the changes in the secretome of the embryonic trophectoderm as it develops during preimplantation, it would be fascinating to learn how it is modulated by the communication with maternal mesenchymal populations via EVs. For that purpose, MSCs have been first stimulated with implantation signals: Activin A, a member of the TGF-ß superfamily, and FSLT1as an activin A inhibitor [97]. FLST1 increase on day 20 uterine flushing and decrease on day 22, according to Kusama et al., whereas activin A elevates on day 20 and increase on day 22. Furthermore, Kusama et al. found that FLST reduced Activin A-induced EMT marker expressions in CT-1 embryonic trophectoderm cells [61]. Activin A or Activin + FLST1 stimulation of embryonic trophectoderm cells with either an epithelial or mesenchymal phenotype resulted in the secretion of implantation proteins (TDGF1, HSPH1, MMP2, and PEG3 [86]) in their EV-cargo [31]. Other authors also found a significant rise in the expression of TDGF1 at day 12, which they linked to the process of invasive growth and embryo implantation [98]. Hatayama et al. found a significant rise in HSPH1 expression in mouse embryos between days 9 and 12, which coincided with organogenesis, and they attributed HSPH1 a role in organogenesis during embryonic development [99]. Yuan et al. later discovered that HSPH1 was present in rat embryos [100]. A substantial expression of HSPH1 in EVs from trophectoderm with the epithelial phenotype was seen in vitro, which was further amplified in the presence of Activin A. HSPH1 expression was much lower in EVs from embryonic trophectoderm with a mesenchymal characteristic [31].

## 5. Challenges and Obstacles in Bovine Embryo-Maternal Communication Study When a Third Viral Actor Breaks in

The main viral pathogens that cause abortion in cattle, sheep, and goats are pestiviruses and herpesviruses [101,102]. However, there is also evidence of naturally occurring abortion and vertical transmission of an *Aphthovirus* in cattle [103].

The genus *Pestivirus* contains the bovine viral diarrhea virus (BVDV), and the *Herpesviridae* family includes the bovine alphaherpesvirus 1 (BoHV-1) and the bovine gammaherpesvirus 4 (BoHV-4). These three viruses attack cattle’s uterus and cause significant economic losses in livestock production [104,105]. Most herds are at risk of infection with bovine viral diarrhea, one of the most common bovine diseases [102].

For laboratory studies, the epithelial cell line Madin Darby Bovine Kidney (MDBK) is commonly used for the in vitro multiplication of BVDV [106,107,108], BoHV-1 [107,109], and BoHV-4 [110,111,112]. In addition, the susceptibility of primary endometrial cultures to BoHV-4 and BVDV has been reported [102,113,114]. Furthermore, BoHV-4 infects both endometrial epithelial cells and endometrial stromal cells [114,115,116]. The infection ability of in vitro-produced embryos by BoHV-1 [117,118,119] and BoHV-4 [111] has also been reported.

The genus *Aphthovirus* within the family *Picornaviridae* contains the foot-and-mouth disease virus (FMDV). During the 2001 outbreak in the United Kingdom, large numbers of abortions were documented within 24 h of the onset of lameness in sheep flocks. In addition, the same FMDV strain that caused the UK 2001 outbreak was later shown to be able to cross the placenta and cause fetal death in newborns under experimental conditions [120,121]. However, there is still a limited description and investigation of FMDV vertical transmission.

The most successful experimental model systems for FMDV isolation, culture, assay, and investigation are cell lines obtained from hamster kidneys (BHK-21) and swine kidneys (PK-15, IB-RS-2, and SK-6) [122]. The relative receptivity of these cell lines to different viral strains varies, leading to virus mutation through virus culture. Furthermore, the FMDV sensitivity of the fetal goat tongue cell line ZZ-R 127 was validated [123]. A primary bovine thyroid (BTY) cell with a high susceptibility to the cow virus was reported [124]. However, primary BTY cells cannot be passaged or frozen without losing their sensitivity. Thus, the preparations of primary BTY cells need the sacrifice of a bovine fetus to get thyroid tissues during the research and diagnostic procedure, which is at odds with the intended aim of “animal welfare” and is both time-consuming and tedious. Mao et al. reported establishing a bovine thyroid cell line (hTERT-BTY) as a tool or an in vitro model to separate, culture, and assay FMDV to study FMDV host-virus interaction [122].

The first reported FMDV receptor, αvβ3, is predominantly expressed in endothelial cells. αvβ6 is highly expressed in the epithelial cells of FMDV target tissues. This is also consistent with the fact that FMDV frequently targets epithelial cells during infection [125]. The reported FMDV receptors include, in addition to the integrin receptors, the heparan sulfate (HS) receptor and a third receptor that has not been yet identified [126].

The role of EVs in viral pathogenesis is being investigated because of their crucial role in mediating intercellular communication. On the one hand, some studies address how EVs are regulated by viruses, focusing on the composition and function of virus-regulated EVs, isolation methods of EVs in the context of virus infection, and prospective antiviral therapies based on EVs utilization. On the other side, researchers are looking at how viral components influence exosome synthesis, composition, and secretion [127,128]. In the context of maternal-embryonic communication, each of the viruses responsible for abortions in cattle (BVDV, BoHV, and FMDV) our knowledge is still incomplete.

Regarding BoHV, the comparable size of smaller EVs and herpesvirus particles, either representing entire enveloped virions in the range of 140–200 nm or non-infectious virus-like particles [129], makes investigations on EVs from herpesvirus-infected cells difficult, restricting the use of several well-known EV separation methods, such as size-exclusion chromatography (SEC). In addition, the herpesvirus assembly and exosome biogenesis pathways may have several common points, as evidenced by human herpesvirus 6 in particular [130]. Exosome formation and alphaherpesvirus virion morphogenesis share certain components of the endosomal sorting complex essential for transport (ESCRT) machinery, like ESCRT-III complex components and Vps4 ATPase [131,132]. Not only is herpesvirus envelope glycoprotein B (gB) an important component of the virus entrance complex, but it is also one of the most well-studied EVs-incorporated viral proteins (for HCMV and HSV-1) [109].

In the case of BVDV, to date there is no information on the implications of EVs in viral pathogenesis, virus-regulated EVs, or possible EVs-isolation methods in the presence of viral particles nor on antiviral therapies based on EVs.

Exosomes produced by FMDV-infected cells have been found to contain FMDV RNAs and most of the viral proteins, enabling productive infection in vitro and in vivo. Furthermore, NAbs have no effects on FMDV infection spread via exosomes. Taken together, the findings suggest that FMDV transmission via exosomes contributes to FMDV’s recognized immunological evasive characteristics [133]. Moreover, FMDV suppresses the secretion of exosomes, suppressing the host cell’s exosome-mediated antiviral immune response [134].

Considering the preceding, it is of great interest to explore the main roles played by mesenchymal stem cells in embryo-maternal communication under healthy conditions in the subsequent stages of preimplantation and embryonic development in which elongating bovine blastocysts are classified as ovoid, tubular, or early filamentous based on their shape and size. Moreover, it is also of significant scientific interest to examine the changes within that communication via EVs, and when virus infection occurs throughout the preimplantation stage.

## Figures and Tables

**Figure 1 cells-11-01858-f001:**
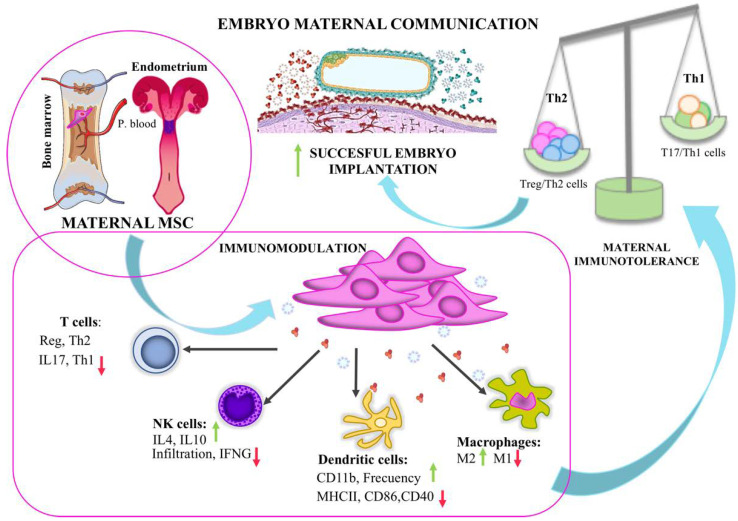
Maternal MSC immunomodulation at the embryo implantation site. The existence of MSC in the reproductive tract is associated with maternal immunoregulation during embryo implantation. MSC exerts immunomodulatory functions in the local environment through cell-to-cell contact or by secreting EV and soluble factors that interact with local immune cell populations, resulting in a shift to a Th2 maternal response.

**Figure 2 cells-11-01858-f002:**
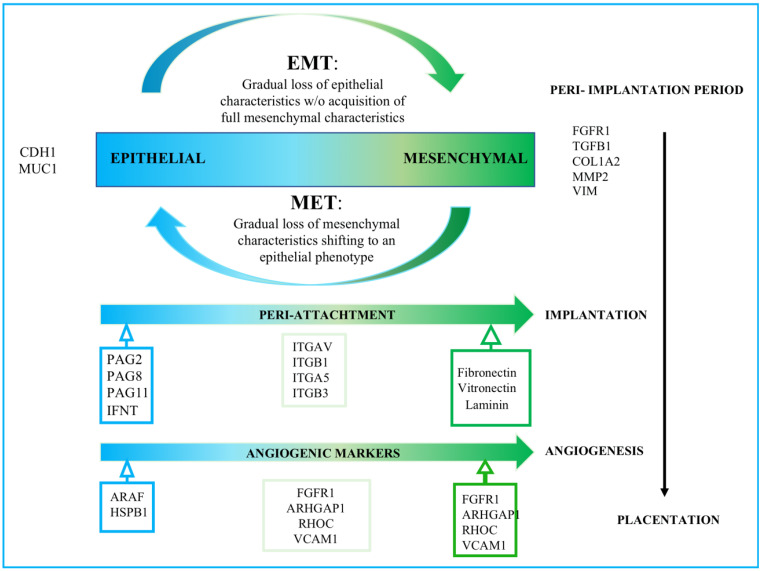
Schematic view of the transdifferentiation cell plasticity of bovine trophectoderm cell populations. Figure based on the characterization of in vitro proteome of soluble fraction and EV from bovine trophectoderm cell lines showing different EMT transitions. Figure adapted from Calle et al. [31].

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
