# Peer review of "Mesenchymal Stem Cells in Embryo-Maternal Communication under Healthy Conditions or Viral Infections: Lessons from a Bovine Model"

_cells, 2022, doi:10.3390/cells11121858_

Round 1

Reviewer 1 Report

1) Chapter 2.3 – I suggest to describe in more details the decidualized cells: characterization, types of cells, their function.

2) Lines 174-176 – The authors should specify the EMT associated genes.

3) Line 217, lines 353-354 and 357-358 – I suggest to rephrase “overrepresented” in order to avoid misunderstanding.

 4) Lines 236-237. The authors should decipher the abbreviation and explain in more details “P4 luteal secretion”.

5) Lines 284-285. Authors should clarify what they mean by " distinct reactions " and add a reference.

6) Lines 292-310. There are no references which confirm the written text.

7) Lines 329-332. Authors should clarify the consequences of MMP2 and PEG3 activation.

8) An obligatory requirement is to add a figure that will reflect all the invasive, proliferative, and immunological tolerance processes in embryo-maternal communication and influence of viral infections on these processes. 

Author Response

Reviewer 1:

1) Chapter 2.3 – I suggest to describe in more details the decidualized cells: characterization, types of cells, their function.

We have rewritten and completed the information on decidualization (Line 110).

“During the window of implantation, after blastocysts attach to the endometrium in both humans and mice, the stromal cells at the implantation site begin to decidualize. [40]. Decidua is identified morphologically as a tissue composed of endometrial fibroblast cells which become rounded or polyhedral, as a result of glycogen or lipid buildup in the cytoplasm. These morphological tissular changes can occur during pregnancy, pseudo-pregnancy, or in experimentally generated deciduomas [41]. One of the earliest uterine adaptations to pregnancy in mice and primates is the differentiation of decidual cells [42] which involves cell proliferation, changes in shape, multinucleation, and the formation of intercellular connections [43]. The process of decidualization, which is required for blastocyst implantation and pregnancy maintenance, involves many genes and signal pathways. Decidualization is not shown in species with noninvasive implantation, such as domestic animals. Ruminants and pigs both have noninvasive implantation, but the type of placentation differs. Epitheliochorial placentation occurs in pig conceptuses, in which the luminal epithelium (LE) is maintained biologically intact throughout pregnancy, and the conceptus trophectoderm only attaches to the apical LE surface without interacting with uterine stromal cells. [44]. Synepitheliochorial placentation in ruminants, on the other hand, results in severe LE erosion owing to syncytium development with trophectoderm binucleate cells. Conceptus tissue opposes but does not penetrate uterine stroma, after Day 19 of pregnancy. According to Johnson et al., the uterine stroma of sheep undergoes a differentiation process similar to decidualization in invasive implanting species, while porcine stroma shows less differentiation than sheep, rodents, or primates. The degree of uterine stromal decidualization thus varies among species and coincides with the depth of trophoblast invasion during implantation [45]. Few studies have focused on the role of alterations in the uterine stroma that support uterine luminal epithelium since domestic animals have mainly  noninvasive implantation.”

2) Lines 174-176 – The authors should specify the EMT associated genes.

Following the reviewer's suggestion, a paragraph was added in chapter 3 (line 186).

“EMT-related genes (SNAI2, ZEB1, ZEB2, TWIST1, TWIST2, and KLF8), as well as cytokeratin, are found in the bovine trophectoderm cells following the conceptus-endometrium attachment [57]”.

3) Line 217, lines 353-354 and 357-358 – I suggest to rephrase “overrepresented” in order to avoid misunderstanding.

The sentences have been rewritten in response to the reviewer's suggestions.

Line 229: “Vimentin and MMP2 expression are significantly increased in mesenchymal trophectoderm cells' EVs and soluble proteins, respectively [31].”

Line 375: “Additionally, VCAM-1 expression is increased in the cargo of mesenchymal trophectoderm cells-EVs. [31].”

Line 377: “Galectin 3 (LGALS3), which also plays important role in immune system regulation by regulating both innate and adaptive responses in physiological and pathological processes [85], has been linked to the implantation process [86] and is overexpressed in mesenchymal trophectoderm cells-EV cargo [31].”

 4) Lines 236-237. The authors should decipher the abbreviation and explain in more details “P4 luteal secretion”.

Following the reviewer's suggestion, the paragraph was rewritten in the 4.2 section.

Line 248: “IFN-τ inhibits luteolysis, which is required for the production of progesterone from the corpus luteum [70].”

5) Lines 284-285. Authors should clarify what they mean by " distinct reactions " and add a reference.

Following the reviewer's suggestion, a paragraph was added in chapter 4.2.

Line 296: “According to the literature, MSCs demonstrate tissue and donor-related diversity, not only in mRNA expression but also in chemokine and cytokine production [78] [79] [80] [81] [82]. The migratory ability of MSC varies depending on the tissue of origin, both in the absence of stimulation and as a result of activation by inflammatory cytokines [83]”.

6) Lines 292-310. There are no references which confirm the written text.

Thank you to the reviewer for pointing up our mistake.

The required bibliographic citations have been included in the paragraph.

Line 312: “Later stages of development are not possible in pure in vitro systems, so investigations of embryo-maternal communication have been based on Trophoblastic cells derived from hatched embryos [30, 31]. Although the diverse trophectoderm cell lines described have different properties, some of them similar to those seen later in embryo development, it would be fascinating to investigate the chemotactic stimuli released by trophoblastic cells derived from elongated embryos. Trophectoderm cells with a mesenchymal phenotype have developmental characteristics that are closer to EMT so would represent a more advanced developmental stage, closer to embryo implantation [31]. A careful analysis was performed, in which the trophectoderm secretome was dissected into soluble mediators and extracellular vesicles (EVs), and the chemotactic activity of both fractions was analyzed independently in parallel. Interestingly, epithelial embryonic trophectoderm lines drive chemotactic migration of maternal eMSCs via both soluble and EVs mediators, while chemoattraction of pbMSCs is induced only via soluble mediators. In contrast, when the embryonic trophectoderm has already developed mesenchymal properties, it induces endometrial or peripheral maternal MSC migration via both soluble and EV-cargo proteins, and these cells can travel greater distances and at faster speeds [31]. Secretome-dependent signaling might thus cause a substantially amplified call effect in MSCs at late implantation stages to guarantee embryo implantation at that key time of pregnancy.”

7) Lines 329-332. Authors should clarify the consequences of MMP2 and PEG3 activation.

Following the reviewer's suggestion, a paragraph was rewritten and another was added in chapter 4.3.

Line 350: “In addition, on Day 13 in vivo conceptuses, MMP2, a member of the matrix metallopeptidase family, which is involved in the degradation of extracellular matrix in normal physiological processes [57], and PEG3 are both up-regulated [57]. MMP2 and PEG3 are essential implantation proteins whose expression has only been observed in in vivo conceptuses. [86].

8) An obligatory requirement is to add a figure that will reflect all the invasive, proliferative, and immunological tolerance processes in embryo-maternal communication and influence of viral infections on these processes. 

To further clarify the reviewer's concerns, we have created two new figures:

“Figure 1. Maternal MSC immunomodulation at the embryo implantation site. The existence of MSC in the reproductive tract is associated with maternal immunoregulation during embryo implantation. MSC exerts immunomodulatory functions in the local environment by cell-to-cell contact, or by secreting EV and soluble factors that interact with local immune cell populations, resulting in a shift to a Th2 maternal response.”

“Figure 2. Schematic view of the transdifferentiation cell plasticity of bovine trophectoderm cell populations. Figure based on the characterization of in vitro proteome of soluble fraction and EV from bovine trophectoderm cell lines showing different EMT transitions. Figure adapted from Calle et al., Int J Mol Sci.,2021.”

Unfortunately, as the editor has indicated, and most likely due to a computer problem, the publisher has been unable to share the graphical abstract that we attached to the article at the time of first submission. Again, we've attached this illustration of a model for bovine mesenchymal stem cells in embryo-maternal communication under normal conditions.

Finally, we think that the data on viral infections and how they impact on embryo-maternal communication are still too scarce to draw a model. We just wanted to highlight in this review article its relevance and the challenges and obstacles that research in this field should overcome in a near future.

Reviewer 2 Report

The present manuscript aimed at reviewing current knowledge regarding the role of MSCs in embryo-maternal communication during normal conditions and following viral infections. The authors managed to gather a significant amount of information and organized it in a logical manner, describing the role of maternal MSC in immunoregulation during implantation as well as their plasticity, followed by a thorough presentation of the role of the secretome in the interaction between embryonic and maternal stem cells.

In order to make this review more appealing to readers, I suggest the following additions:

- graphical representations of the various mechanisms that are described throughout the manuscript and tables summarizing findings of various authors; they are absolutely necessary for a review paper, as they provide a quick insight into the intimate mechanisms that are detailed in the text. 

- a general conclusion of the review, which should also point out future research trends.

- the whole manuscript needs proper English language editing, since there are still several grammar mistakes.

Author Response

Reviewer 2:

The present manuscript aimed at reviewing current knowledge regarding the role of MSCs in embryo-maternal communication during normal conditions and following viral infections. The authors managed to gather a significant amount of information and organized it in a logical manner, describing the role of maternal MSC in immunoregulation during implantation as well as their plasticity, followed by a thorough presentation of the role of the secretome in the interaction between embryonic and maternal stem cells.

In order to make this review more appealing to readers, I suggest the following additions:

- graphical representations of the various mechanisms that are described throughout the manuscript and tables summarizing findings of various authors; they are absolutely necessary for a review paper, as they provide a quick insight into the intimate mechanisms that are detailed in the text.

Unfortunately, as the editor has indicated, and most likely due to a computer problem, the publisher has been unable to share the graphical abstract that we attached to the article at the time of submission with the reviewers.

We've attached this illustration of a model for bovine mesenchymal stem cells in embryo-maternal communication under normal conditions.

In addition, we have included two new figures:

“Figure 1. Maternal MSC immunomodulation at the embryo implantation site. The existence of MSC in the reproductive tract is associated with maternal immunoregulation during embryo implantation. MSC exerts immunomodulatory functions in the local environment by cell-to-cell contact, or by secreting EV and soluble factors that interact with local immune cell populations, resulting in a shift to a Th2 maternal response.”

“Figure 2. Schematic view of the transdifferentiation cell plasticity of bovine trophectoderm cell populations. Figure based on the characterization of in vitro proteome of soluble fraction and EV from bovine trophectoderm cell lines showing different EMT transitions. Figure adapted from Calle et al., Int J Mol Sci.,2021.”

- a general conclusion of the review, which should also point out future research trends.

Following the reviewer's suggestion, a paragraph was rewritten (Line 484).

“Considering the foregoing, it is of great interest to explore the main roles played by mesenchymal stem cells in embryo-maternal communication under healthy conditions in the subsequent stages of pre-implantation and  embryonic development, in which elongating bovine blastocysts are classified as ovoid, tubular, or early filamentous based on their shape and size. Moreover, it is also of significant scientific interest to examine the changes within that communication via EVs and when virus infection occurs throughout the pre-implantation stage.”

- the whole manuscript needs proper English language editing, since there are still several grammar mistakes.

The entire document has been verified for grammatical and linguistic errors.

Round 2

Reviewer 2 Report

The authors made the required changes and therefore I agree with the publication of the current form of the manuscript.